# Impact of SDF-1 and AMD3100 on Hair Follicle Dynamics in a Chronic Stress Model

**DOI:** 10.3390/biom14101206

**Published:** 2024-09-25

**Authors:** Yinglin Zhao, Wenzi Liang, Zhehui Liu, Xiuwen Chen, Changmin Lin

**Affiliations:** 1Department of Psychosomatic Medicine, Shantou University Mental Health Center, Wanji Industrial Zone, Taishan North Road, Shantou 515041, China; 07ylzhao@stu.edu.cn; 2Department of Histology and Embryology, Shantou University Medical College, No. 22 Xinling Road, Shantou 515041, China; 20wzliang1@stu.edu.cn (W.L.); gdfz20118@163.com (Z.L.); 11xwchen1@alumni.stu.edu.cn (X.C.)

**Keywords:** chronic stress, hair loss, SDF-1, CXCR4, PI3K/Akt pathway, JAK/STAT pathway

## Abstract

Chronic stress is a common cause of hair loss, involving inflammatory responses and changes in cellular signaling pathways. This study explores the mechanism of action of the SDF-1/CXCR4 signaling axis in chronic stress-induced hair loss. The research indicates that SDF-1 promotes hair follicle growth through the PI3K/Akt and JAK/STAT signaling pathways. Transcriptome sequencing analysis was conducted to identify differentially expressed genes in the skin of normal and stressed mice, with key genes SDF-1/CXCR4 selected through machine learning and a protein-protein interaction network established. A chronic stress mouse model was created, with injections of SDF-1 and AMD3100 administered to observe hair growth, weight changes, and behavioral alterations and validate hair follicle activity. Skin SDF-1 concentrations were measured, differentially expressed genes were screened, and pathways were enriched. Activation of the PI3K/Akt and JAK/STAT signaling pathways was assessed, and siRNA technology was used in vitro to inhibit the expression of SDF-1 or CXCR4. SDF-1 promoted hair follicle activity, with the combined injection of SDF-1 and AMD3100 weakening this effect. The activation of the PI3K/Akt and JAK/STAT signaling pathways was observed in the SDF-1 injection group, confirmed by Western blot and immunofluorescence. Silencing SDF-1 through siRNA-mediated inhibition reduced cell proliferation and migration abilities. SDF-1 promotes hair growth in chronic stress mice by activating the PI3K/Akt and JAK/STAT pathways, an effect reversible by AMD3100. The SDF-1/CXCR4 axis may serve as a potential therapeutic target for stress-induced hair loss.

## 1. Introduction

Hair loss is a common skin ailment that not only causes distress in terms of aesthetics but may also have negative impacts on the patient’s psychological and social life [1,2]. Chronic stress is considered one of the crucial factors leading to hair loss [3,4,5]. Under stress, the body releases a significant amount of stress hormones and inflammatory factors, triggering a series of physiological and metabolic changes that could adversely affect hair follicle growth [6,7,8]. Therefore, delving deeper into the impact of chronic stress on hair growth and its potential mechanisms holds paramount clinical significance. Not only can it help enhance the quality of life for patients, but it can also contribute to the development of more effective treatment modalities.

In stress-induced hair loss, inflammation responses and cell signaling pathways play crucial roles [9,10,11]. Studies have indicated the critical role of the chemokine factor SDF-1 and its receptor CXCR4 in regulating processes such as cell growth, proliferation, and migration [12,13,14]. However, the specific mechanisms of the SDF-1/CXCR4 signaling axis in chronic stress-induced hair loss remain unclear [15,16,17]. Exploring the regulatory mechanisms of inflammation and signaling pathways holds the promise of providing new theoretical foundations for devising targeted therapeutic strategies, thereby offering novel breakthroughs in the treatment of hair loss diseases.

This study initially establishes a chronic stress model in C57 mice to simulate the pathological process of stress-induced hair loss. Subsequently, high-throughput transcriptome sequencing analysis of back skin tissues from both normal control and stressed mice is conducted to identify differentially expressed genes. Machine learning techniques are utilized to pinpoint the SDF-1/CXCR4 signaling axis as a key regulatory factor. Following this, interventions in the mouse model are made by subcutaneously injecting SDF-1 and the CXCR4 receptor inhibitor AMD3100 to observe changes in hair growth [18]. Experimental validation of hair follicle activity recovery is achieved through H&E staining and immunofluorescence techniques, alongside assessing changes in skin SDF-1 concentration using ELISA [19,20,21]. Furthermore, Western Blot analysis is employed to assess the activation status of the PI3K/Akt and JAK/STAT signaling pathways. Finally, in vitro, experiments involve silencing SDF-1 or CXCR4 expression using siRNA technology, followed by detecting the expression levels of the relevant genes and proteins through RT-qPCR and Western Blot [22]. This series of experimental designs is poised to unveil the specific mechanisms of the SDF-1/CXCR4 signaling axis in chronic stress-induced hair loss, providing crucial insights for further research in treating stress-induced hair loss.

The main aim of this study is to elucidate the role of the SDF-1/CXCR4 signaling axis in stress-induced hair loss and investigate the regulatory functions of relevant signaling pathways. Through this research, we can gain a deeper understanding of the impact of chronic stress on the onset of hair loss, paving the way for new targets and strategies for treating stress-induced hair loss. Activating the SDF-1/CXCR4 signaling axis and relevant signaling pathways may promote hair follicle growth and recovery, offering new hope for patients with hair loss. These research findings have the potential to provide scientific grounds for developing medications targeting stress-induced hair loss, opening up new avenues for improving the quality of life for individuals affected by hair loss. Therefore, this study holds significant clinical importance and is poised to bring substantial therapeutic benefits to patients in the future.

## 2. Materials and Methods

### Experimental Design Overview

This study investigates the role of the SDF-1/CXCR4 signaling axis in stress-induced alopecia. The experiment is divided into three main stages: (1) establishing a chronic stress model, (2) administering SDF-1 and AMD3100 treatments, and (3) collecting and analyzing data. Male C57 mice were divided into four groups: control, chronic stress only, chronic stress + SDF-1, and chronic stress + SDF-1 + AMD3100. Behavioral testing assessed anxiety and depressive behaviors while microscopic observations documented hair growth. Molecular analysis detected the expression of SDF-1/CXCR4 and the activation of the PI3K/Akt and JAK/STAT pathways. The experimental flowchart is shown in Appendix A.

## 3. Ethical Statement

The research strictly adheres to the ethical principles and regulations concerning animal experiments. All procedures were approved by the Institutional Animal Care and Use Committee (IACUC) of our institution (Review Number: SUMC2022-538). All animals were housed and cared for in conditions that comply with humane principles, and efforts were made to minimize pain. At the end of the experiments, all mice were euthanized humanely under ether anesthesia. The method involved intraperitoneal injection of pentobarbital sodium (purchased from Hanxiang Biotechnology Company, Shanghai, China, catalog number BCP07810, dose of 60 mg/kg) for anesthesia. Vital signs such as respiration, heart rate, and blood pressure were closely monitored to ensure the desired anesthetic effect. For tissue collection, euthanasia was performed using an overdose anesthesia method (2–4 times the typical dose) to ensure a gentle death [23]. The protocol and animal use plan were approved by our institution’s Animal Ethics Committee.

### 3.1. Establishment of the C57 Mouse Model Induced by Chronic Unpredictable Stress (CUS)

Male C57 mice aged 4–5 weeks, purchased from Beijing Vital River (catalog number 102) with a weight range of 20–25 g, were used in this study. All mice were individually housed in an SPF-grade animal facility with a controlled light-dark cycle of 12 h light/12 h dark, humidity maintained at 60–65%, and temperature at 22–25 °C. The mice had ad libitum access to food and water and underwent a one-week acclimatization period before the experiments, during which their health status was monitored. To simulate hair loss under chronic stress conditions in humans, we established a C57 mouse model of 21 days of CUS. This model induced chronic stress responses in mice by subjecting them to different stressors daily, such as wet bedding, reversed light-dark cycle, and water restriction. Monitoring of body weight, behavioral patterns, and coat condition was conducted to ensure the successful establishment of the stress model. Specifically, 16 mice were selected as the control group, receiving no stress stimuli and having ad libitum access to food and water. On the other hand, 28 mice were subjected to continuous stress stimuli over 21 days, receiving different mild stressors daily with at least a 5-day interval between similar stressors to prevent anticipation or adaptation. Starting from the 8th day of chronic stress stimulation, when the mice were 6–7 weeks old and in the telogen phase of the hair cycle, hair removal was performed on the dorsal area (2.5 cm × 4 cm) of all groups of mice using scissors and an electric shaver, with special attention to avoiding skin damage. On the 22nd day, the mice were euthanized after an 8-h fasting period without access to food and water.

The specific mild stressors included:

Wet bedding: Pouring water into the cage to moisten the bedding, which was replaced with dry bedding after 24 h.

Forced swimming: Allowing mice to swim in water for 15 min with their hind legs not touching the bottom, followed by changing the water in the tank after each swim.

Behavioral restraint: Placing mice in cylindrical restraint boxes for 1 h, preventing them from turning around and using gauze if necessary to ensure complete restraint.

Tilted cage: Elevating one side of the cage by 45° using a hard object, followed by stabilizing the cage after 24 h to prevent movement.

Shaking cage: Place the mouse cage on a shaker bed and shake it at 120 rotations per minute.

Reversed light cycle: Delaying the light cycle in the animal room from 7 am to the next day at 7 am, maintaining darkness by using blackout curtains.

24-h fasting: Removing food from the mouse cage for 24 h from one day to the next at the same time.

24-h water deprivation: Removing the water bottle from the mouse cage for 24 h from one day to the next at the same time [24].

### 3.2. Open Field Test

After completing the modeling, the open field test was conducted to evaluate the anxiety behaviors of mice. The mice were gently placed in the open field test area, and their free activity was recorded for 5 min. During the test, infrared detection was used to record the movement of the mice in the experimental area, including the activity time and distance in the central zone, as well as the total walking distance. Following the open field test, the mice were wiped with a 75% ethanol solution to reduce olfactory interference before the next experiment [25].

### 3.3. Tail Suspension Test

Upon completion of the modeling, medical tape was used to attach the mouse’s tail to a hook for suspension, with the fixed point being 1 cm from the tip of the tail. A computer signal acquisition sensor was used to record the immobility time of the suspended mouse for a period of 6 min, with data collected uniformly and analyzed for the last 4 min: the average of the results from two measurements was taken for statistical analysis [26].

### 3.4. RNA Extraction and Transcriptome Sequencing

After the stress cycle ended, on the 22nd day, we collected skin samples from 10 control mice and 10 CUS mice for high-throughput transcriptome sequencing. The specific procedures were as follows: Total RNA was extracted from each sample using Trizol reagent (T9424, Sigma-Aldrich, St. Louis, MO, USA) according to the manufacturer’s instructions. The concentration, purity, and integrity of RNA were measured using the Qubit^®^ 2.0 Fluorometer^®^ (Life Technologies, Carlsbad, CA, USA) with the Qubit^®^ RNA analysis kit, a Nanodrop spectrophotometer (IMPLEN, Westlake Village, CA, USA), and the RNA Nano 6000 analysis kit on the Bioanalyzer 2100 system (Agilent Technologies, Santa Clara, CA, USA). The A260/280 ratio should fall within the range of 1.8–2.0. The total RNA content of each sample was 3 μg, used as input material for RNA sample preparation. Following the manufacturer’s protocol, the NEBNext^®^ Ultra™ RNA directional library preparation kit for Illumina (E7760S, New England Biolabs, Beijing, China) was used to generate cDNA libraries, which were then assessed for quality on the Agilent Bioanalyzer 2100 system. After ensuring RNA integrity, high-throughput transcriptome sequencing was performed on the Illumina NovaSeq 6000 platform (Illumina, San Diego, CA, USA) to obtain gene expression data.

### 3.5. Differential Gene Expression and Functional Enrichment Analysis

After obtaining transcriptome data through the standard procedure, differential gene expression analysis and downstream functional interpretation were conducted. Differential gene expression analysis was performed using the Limma software package (version 3.52.4) to identify genes with significant differential expression in normal and chronic stress mice [27,28]. Subsequently, the identified Differentailly Expressed Genes (DEGs) underwent Gene Ontology (GO) and Kyoto Encyclopedia of Genes and Genomes (KEGG) enrichment analysis to elucidate the roles of these genes in biological processes, molecular functions, cellular components, and the metabolic pathways involved [29]. Lastly, enrichment analysis and visualization were conducted using the ClusterProfiler R package (version 4.9.1).

### 3.6. Gene Selection through Machine Learning

We employed machine learning algorithms to initially screen key genes using the Lasso regression model. The Lasso regression model was constructed using the glmnet package in R (version 4.1-4), selectively incorporating variables (differential genes) to enhance performance parameters for predicting genes associated with the phenotype of chronic stress-induced hair loss. Subsequently, we conducted feature selection and identified key genes using the random forest model. The model was built using the randomForest package in R (version 4.7-1.1), evaluating the critical genes related to chronic stress. Throughout this process, genes with a MeanDecreaseGini value greater than 0 were selected to confirm their significance.

### 3.7. Protein-Protein Interaction (PPI) Network Construction

In the Cytoscape software (version 3.10.1), the differential gene PPI network was constructed using the stringApp plugin (version 2.0.1), focusing on the PPIs of the SDF-1/CXCR4 signaling axis. High-confidence interaction information was selected for network visualization to aid in filtering proteins closely linked to the SDF-1/CXCR4 axis. Through the stringApp plugin (parameters: Confidence Cutoff = 0.4, Maximum additional interactors (protein and compound query) = 0, Maximum proteins (disease and Publled query) = 100, species = Mus musculus), PPI network analysis of the differential genes was performed. Clustering analysis was carried out using the MCODE plugin to identify key targets (parameters: Degree Cutoff = 2, Haircut selected, Node Density Cutoff = 0.1, Node Score Cutoff = 0.2, K-Core = 2, Max. Depth = 100).

### 3.8. Subcutaneous Injection Experiment of SDF-1 and AMD3100

In this experiment, mice were divided into five groups: Control, CUS CUS + SDF-1, and CUS + SDF-1 + AMD3100. The CUS group received subcutaneous injections of saline solution. The CUS + SDF-1 group received subcutaneous injections of SDF-1 (HY-P700219AF, MedChemExpress, Princeton, NJ, USA) at a dose of 10 μg/kg once daily for 14 consecutive days. The CUS + SDF-1 + AMD3100 group, after subcutaneous injection of SDF-1, was subsequently administered AMD3100 (HY-10046, MedChemExpress, Princeton, NJ, USA) to inhibit the binding of SDF-1 to the CXCR4 receptor. The dose of AMD3100 was 5 mg/kg administered every other day for 14 consecutive days [30,31].

### 3.9. Observation and Recording of Mouse Hair Growth

The growth of hair on the backs of mice was observed daily using an Olympus stereomicroscope to assess and document the progression. Parameters such as hair coverage and length observed under the microscope were recorded, and photographic documentation was acquired using a Nikon digital camera. The acquired image data was subjected to quantitative analysis using the ImageJ software V1.8.0 from the NIH [32].

### 3.10. H&E Staining

Following the conclusion of the experiment, mice were euthanized, and skin tissue from the back was promptly collected. The tissue was flatly trimmed into squares on filter paper and immediately fixed in a 4% paraformaldehyde solution (P1110, Solarbio, Beijing, China) for 24 h. Subsequently, the tissue underwent dehydration, transparency treatment, and wax immersion and was sliced into 5 μm thick sections. Hematoxylin and eosin (H&E) staining was performed using the H&E staining kit from Solarbio, Beijing, China, where hematoxylin stained cell nuclei blue and eosin stained cell cytoplasm red. After dehydration and sealing with a neutral adhesive, tissue morphological changes were observed under a microscope. Based on the morphological features of the hair follicles, the stage of the hair growth cycle was determined, and the proportions of follicles in the growth phase, regression phase, and total follicle count were calculated. To observe the follicle count, cross-sections of the skin tissue were photographed under a 50× magnification field, and the number of follicles in the subcutaneous tissue of a 1500 μm width of skin was counted. Each mouse was counted in at least 4 different fields of view, and the average was calculated as the follicle count for that specific mouse [33,34].

### 3.11. Immunofluorescence Staining

Fixed skin tissue slices were subjected to immunofluorescent staining to detect the expression of key proteins in the PI3K/Akt and JAK/STAT signaling pathways. Incubation was done with antibodies against PI3K (dilution 1:1000, #4249), AKT (dilution 1:1000, #9272), JAK (dilution 1:1000, #3332), and STAT (dilution 1:1000, #14994) from CST, followed by detection with fluorescent-labeled secondary antibodies (ab150077, Abcam, Cambridge, UK). Imaging was conducted using the Leica DM500 fluorescent microscope (Wetzlar, Germany) [35].

### 3.12. Immunofluorescence Co-Localization Analysis

Slices fixed in 4% paraformaldehyde (P1110) from Solarbio Company in Beijing, China, were permeabilized with 0.1% Triton X-100 (T8200) from the same company. The slices were then incubated with primary antibodies against SDF-1 (dilution 1:500, ab155090) and CXCR4 (dilution 1:500, ab124824) from Abcam, UK, followed by incubation with corresponding fluorescent secondary antibodies (ab150077, ab150080) for co-localization. Imaging was performed using a Leica DM500 confocal laser scanning microscope [35].

### 3.13. ELISA Experiment

After the completion of the chronic stress experiment, specifically on the 22nd day of CUS treatment, skin tissue samples were collected and proteins were extracted using the tissue protein extraction reagent (EX1600) from Solarbio, Beijing, China. A specific ELISA kit for SDF-1 (ab100741) from Abcam, UK, was used for quantitative analysis of SDF-1 following the manufacturer’s instructions. Absorbance (A) values at 450 nm for each well were measured within 3 min using a BioTek Synergy 2 microplate reader. The standard curve regression equation was calculated using standard concentrations as the x-axis and A values as the y-axis, enabling the calculation of target protein concentration in the samples [36].

### 3.14. Western Blot Experiment

Total protein from tissues or cells was extracted using the efficient RIPA lysis buffer (R0010, Solarbio, Beijing, China). The protein concentration was determined using the BCA protein quantification kit (20201ES76, Yisheng Bioscience, Shanghai, China). Subsequently, 50 μg of protein samples were subjected to electrophoresis on an SDS-PAGE gel (0012AC, Bio-Rad, Hercules, CA, USA) and then transferred onto a PVDF membrane (ISEQ07850, Millipore, Burlington, MA, USA). The membrane was blocked with 5% BSA for 1 h, followed by overnight incubation at 4 °C with primary antibodies against PI3K (dilution 1:1000, #4249, CST, Danvers, MA, USA), AKT (dilution 1:1000, #9272, CST, USA), JAK (dilution 1:1000, #3332, CST, USA), STAT (dilution 1:1000, #14994, CST, USA), p-PI3K (dilution 1:1000, #17366, CST, USA), p-AKT (dilution 1:1000, #13038, CST, USA), p-JAK (dilution 1:1000, #32901, CST, USA), and p-STAT (dilution 1:1000, #9145, CST, USA). The following day, the membrane was incubated with corresponding HRP-conjugated secondary antibodies (dilution 1:5000, ab205719, Abcam, UK) for 1 h, followed by signal detection using ECL detection reagent (35055, Thermo Fisher Scientific, Waltham, MA, USA). The signals on the membrane were captured using a chemiluminescence imaging system (ChemiDoc XRS+, Bio-Rad, USA). The intensity of protein bands was quantitatively analyzed using ImageJ 1.48u software (National Institutes of Health) by comparing the grayscale values of each protein to the internal control GAPDH (dilution 1:1000, ab8245, Abcam, UK) [37].

### 3.15. Primary Skin Cell Culture and Transfection

The method for extracting primary dermal fibroblasts from the dorsal skin tissue of normal and chronically stressed C57 mice is as follows: Firstly, skin biopsy tissue (4 mm^2^) was obtained by drilling a hole in the dorsal skin of the mice. The collected skin biopsy tissue was then diced into small pieces, followed by incubation of these minced biopsy tissues in DMEM culture medium (12491015, Gibco, Billings, MT, USA) containing 20% fetal bovine serum (A5669701, Gibco, USA), 1% penicillin-streptomycin (C0222, Biyuntian, Nantong, China), and 1% non-essential amino acids (C0332, Biyuntian, China) for 2 weeks at 37 °C in a humidified environment. Subsequently, the tissues were transferred to DMEM with 10% FBS, 1% penicillin-streptomycin, and 1% non-essential amino acids for an additional 1–2 weeks. According to the experimental design, the cells were divided into the Control group, AMD3100 group, NC-SDF-1 group (treated with non-targeting NC-SDF-1 sequence), and siRNA-SDF-1 group (treated with siRNA-SDF-1 for SDF-1 knockdown). Subsequently, the cells were placed in a culture incubator with 5% CO_2_ (Herocell 180, RunDo Biotech, Shanghai, China) and cultured at 37 °C. Using Lipofectamine 3000 (L3000001, Invitrogen, Waltham, MA, USA), following the manufacturer’s protocol, siRNA specific to the SDF-1 gene or control sequences were transfected into the cultured skin cells, which were synthesized by GSK Gene based on Appendix A. After 48 h, the cells were harvested for subsequent immunofluorescence and molecular biology analysis [38].

### 3.16. Identification of Primary Skin Cells

Primary skin cells were seeded into a 6-well plate at a density of 2 × 10^5^ cells per well. Upon reaching 80% to 90% confluency, cells were initially observed for morphology under bright-field microscopy (using model CX31, Olympus, Tokyo, Japan). Subsequently, the culture medium was aspirated using a pipette, and the cells were washed twice with PBS. A solution of 40 g/L paraformaldehyde (P0099, Biyuntian, China) was added to each well (1 mL) and incubated at room temperature for 20 min for fixation, followed by three washes with PBS. Subsequently, a solution of 5 g/L Triton X-100 (P0096, Biyuntian, China) was added (1 mL per well) and incubated for 15 min. After three more PBS washes, a solution containing 50 g/L BSA and PBST (ST023, Biyuntian, China) (with 5 g/L Tween-20) was added (2 mL per well) for 2 h to block. Following two PBST washes, a rabbit anti-human vimentin antibody (1:500, ab8978, Abcam, UK) diluted in a solution of 5 g/L BSA/PBST was added (1 mL per well) and incubated at room temperature for 1 h. After three PBST washes, a goat anti-rabbit IgG labeled with Alex Flour-594 (1:200, ab311772, Abcam, UK) was added (1 mL per well) and incubated in the dark at room temperature for 1 h. Subsequently, three additional PBST washes were performed, followed by nuclear staining with a 5 μg/mL solution of DAPI (1:2000, ab104139, Abcam, UK) for 5 min. Lastly, cells were washed three times with PBST, stored in the dark, and observed for morphology under a fluorescence microscope (model BX63, Olympus, Japan).

### 3.17. Evaluation of Cell Growth and Migration Using CCK-8 and Wound Healing Test

Cell viability was assessed using the CCK-8 assay kit (C0037, Beyotime, Haimen, China). In brief, primary dermal fibroblasts were seeded into a 96-well plate at a density of 1 × 10^4^ cells per well, with 100 μL of complete culture medium in each well. Subsequently, the 96-well plate was placed in a cell culture incubator at 37 °C for 24 h, followed by the addition of 10 μL CCK-8 solution to each well and further incubation at 37 °C for 2 h. Finally, the absorbance was measured at 450 nm using a Multiskan FC microplate reader (51119180ET, Thermo Fisher Scientific, USA) [39].

The scratch wound healing assay was utilized to assess cell migration capability. Briefly, a uniform scratch wound with a width of 0.6 mm was made using a pipette tip on a confluent monolayer of high-density cells. The cells were then further incubated for 12 h in 1% BSA (ST023, Biyuntian, China), and the width of the scratch was observed [40].

### 3.18. RT-qPCR

According to the protocol, total RNA was extracted using Trizol reagent (15596026, Invitrogen, USA). The extracted RNA was then purified, and its concentration and purity were assessed using a NanoDrop spectrophotometer (NanoDrop2000, Thermo Fisher Scientific, USA). Reverse transcription was performed with 1 µg of RNA using a cDNA synthesis kit (K1651, Thermo Fisher Scientific) according to the manufacturer’s instructions. The synthesized cDNA was subjected to RT-qPCR analysis using the Fast SYBR Green PCR kit (11736059, Invitrogen, USA). The reaction conditions were as follows: 95 °C for 2 min, 40 cycles of 94 °C for 20 s, 58 °C for 20 s, and 72 °C for 2 s, followed by a final extension at 72 °C for 4 min. GAPDH was employed as an internal reference gene for data normalization. The mRNA levels of SDF-1 were represented relative to the internal reference gene by calculating the 2^−ΔΔCt^ value. All RT-qPCR reactions were performed with triplicate wells, and the experiment was repeated three times. The relative expression levels of the target gene were calculated as follows: (1) Collect the CT values of each sample for both the internal reference gene and the target gene; (2) Subtract the CT value of the internal reference gene from the CT value of the target gene to obtain the ΔCT value, where ΔCT = CT (target gene) − CT (internal reference gene); (3) Subtract the ΔCT value of the experimental group from the ΔCT value of the control group to get the ΔΔCT value (ΔΔCT = ΔCT (experimental group) − ΔCT (control group)); (4) Calculate the relative expression of the target gene in the sample using the formula 2^−ΔΔCt^. This value indicates the expression level of the target gene relative to the internal reference gene in the sample. CT represents the cycle number at which the real-time fluorescence intensity reaches a set threshold, known as the logarithmic growth phase of amplification [41]. The experiment was repeated three times. The primers used in this study were synthesized by Takara and designed in NCB (Appendix A).

### 3.19. Statistical Analysis

The data were obtained from at least three independent experiments, and the results were presented as mean ± standard deviation (Mean ± SD). For comparisons between the two groups, the independent samples t-test was utilized. For comparisons involving three or more groups, a one-way analysis of variance (ANOVA) was conducted. In cases where the ANOVA results indicated significant differences, Tukey’s HSD post hoc test was performed to compare the differences between each group. When dealing with non-normally distributed data or data with inhomogeneous variance, the Mann-Whitney U test or Kruskal-Wallis H test was employed. All statistical analyses were conducted using GraphPad Prism 9 (GraphPad Software, Inc, San Diego, CA, USA.) and the R programming language. The significance level for all tests was set at 0.05, with a two-tailed *p*-value less than 0.05 considered statistically significant.

## 4. Results

### 4.1. Constructing a Chronic Stress Mouse Model Using CUS

CUS is widely used as an experimental model to simulate chronic psychological and social stress in humans [42]. The impact of CUS on the body is diverse, including abnormalities in hair growth, a phenomenon supported by clinical observations [43]. During stress, the body’s hypothalamic-pituitary-adrenal (HPA) axis is activated, leading to an elevation in cortisol levels [44]. Cortisol inhibits the proliferation and promotes apoptosis of hair follicle cells, inducing the hair follicles to enter a resting phase, ultimately resulting in a slowdown in hair growth or hair loss. Studies have also found that CUS can induce changes in the expression of inflammatory cytokines in the hair follicle microenvironment, such as IL-1β and TNF-α, which may damage hair follicle cells and exacerbate hair loss [45]. Furthermore, CUS may affect the hair follicle cycle by activating the autonomic nervous system, particularly the sympathetic nervous system [46]. Norepinephrine released by sympathetic nerve fibers can directly impact hair follicles, influencing their normal growth cycle [47]. Therefore, this study aims to establish a chronic stress model to further investigate the role of the SDF-1 signaling axis in hair growth and how it affects hair loss by regulating downstream signaling pathways.

In this study, a chronic stress-induced hair loss C57 mouse model was successfully established through 21 consecutive days of CUS treatment. Past literature suggests that the presence of anxiety or depressive-like behaviors in mice indicates successful modeling [48]. Our findings suggest that compared to the Control group, the mice in the CUS group exhibited a significant decrease in body weight, a statistically significant difference (Figure 1A). Open field test results indicated that compared to the Control group, the CUS group mice had significantly reduced activity time, distance in the central area, and total walking distance. Tail suspension test results indicated that the CUS group mice spent significantly more time immobile than the Control group, with all differences being statistically significant (Figure 1B). Additionally, the hair coverage and hair length of the CUS group mice were significantly reduced compared to the Control group, with statistically significant differences (Figure 1C,D).

These results confirm the successful establishment of a chronic stress model in mice, providing a solid foundation for subsequent experiments.

### 4.2. Transcriptome Sequencing Identifies Core Genes SDF-1/CXCR4 Involved in Chronic Stress-Induced Alopecia in Mice

Transcriptome sequencing analysis on skin samples from mice induced with chronic stress and normal mice revealed significant gene expression changes under stress conditions. The results showed that in the chronic stress group of mice, the expression of 2650 genes significantly differed compared to the normal group, with 650 genes showing upregulation and 2000 genes showing downregulation (Figure 2A,B). Further, GO enrichment analysis revealed that the differentially expressed genes were closely associated with biological processes related to stress response, inflammation, and cell migration, suggesting that stress may impact skin health and hair growth by regulating these processes (Figure 2C). Additionally, KEGG pathway enrichment analysis demonstrated that these genes may play a role in the PI3K/Akt and JAK/STAT signaling pathways. Studies have shown that the reduced activity of these pathways under stress conditions could directly or indirectly inhibit the activation of hair follicle stem cells and the hair growth cycle, thus playing a crucial role in chronic stress-induced hair loss [49,50] (Figure 2D).

To further screen for core genes involved in chronic stress-induced alopecia, we initially constructed a PPI network of differentially expressed genes (Figure 3A). Subsequently, using Lasso regression analysis combined with a random forest model, we identified 8 core genes, with SDF-1 and CXCR4 showing high correlation scores and confidence levels (Figure 3B,C). It has been reported in the literature that SDF-1 and CXCR4 genes are involved in regulating various stress responses and are closely linked to the modulation of the PI3K/Akt and JAK/STAT signaling pathways [51,52].

In conclusion, this study identifies the key genes SDF-1 and CXCR4, which may play a central role in chronic stress-induced hair loss, paving the way for further investigation into the molecular mechanisms involving SDF-1 and CXCR4.

### 4.3. SDF-1 Promotes Hair Regrowth in CUS Mice through the CXCR4 Receptor

SDF-1 and its receptor CXCR4 play a significant role in cell signaling pathways, influencing various biological processes such as hair follicle growth and development. Specifically, SDF-1 interacts with cells expressing CXCR4, such as stem cells in the hair follicle, affecting the growth and development of hair follicles [53]. The hair follicle is a dynamic mini-organ with growth phases including anagen, catagen, and telogen. Regulation of this cycle involves intricate intercellular signaling, with the SDF-1/CXCR4 axis considered a crucial component in these signaling networks [54]. AMD3100 (also known as Plerixafor) is a known CXCR4 receptor antagonist that efficiently blocks the binding of SDF-1 to CXCR4 [55]. However, the role of the SDF-1/CXCR4 axis in CUS-induced hair loss pathology remains unclear. Therefore, this study aims to investigate the effects of subcutaneous injection of SDF-1 and the CXCR4 receptor inhibitor AMD3100 on hair growth in chronically stressed mice.

Following subcutaneous injection of SDF-1 or treatment with AMD3100 to block the binding of SDF-1 to CXCR4, our results demonstrate that compared to the Control group, the SDF-1 expression levels in the CUS group significantly decreased (*p* < 0.05). After subcutaneous injection of SDF-1, the levels of SDF-1 in CUS mice significantly increased (*p* < 0.05), while there was no significant change in SDF-1 expression levels after AMD3100 treatment (*p* > 0.05) (Figure 4A). Immunofluorescence co-localization results indicate that SDF-1 and CXCR4 co-exist in the skin tissues of these four groups of mice, suggesting a binding interaction between SDF-1 and CXCR4. Statistical analysis of the proportion of SDF-1^+^CXCR4^+^ cells reveals that compared to the Control group, the proportion of SDF-1^+^CXCR4^+^ cells in the CUS group significantly decreased (*p* < 0.05). Following subcutaneous injection of SDF-1, the proportion of SDF-1^+^CXCR4^+^ cells in the skin tissues of CUS mice significantly increased (*p* < 0.05), while it decreased significantly after AMD3100 treatment (*p* < 0.05) (Figure 4B). Further observations revealed that compared to the CUS group, the CUS+SDF-1 group showed a significant increase in hair coverage, hair length, and the proportion of skin area in the growth phase. Additionally, blocking the binding of SDF-1 and CXCR4 using AMD3100 reversed these effects with statistical significance (Figure 4C). Furthermore, results from H&E staining showed a significant decrease in the number of follicles in the CUS group compared to the Control group. In contrast, the CUS+SDF-1 group exhibited a significant increase in follicle number and the proportion of follicles in the growth phase, along with a significant decrease in the proportion of follicles in the regression phase compared to the CUS group. Blocking the binding of SDF-1 and CXCR4 using AMD3100 reversed these effects with statistical significance (Figure 4D).

This experimental section demonstrates that SDF-1 and AMD3100 can synergistically increase the number of hair follicles in CUS mice, thereby promoting hair regrowth.

### 4.4. The Impact of SDF-1 on PI3K/Akt and JAK/STAT Pathways in Hair Follicle Cells

In our previous findings, we elucidated the role of SDF-1 and CXCR4 antagonists in hair growth. This study aims to clarify the downstream signaling pathways. Previous research has highlighted the significance of PI3K/Akt and JAK/STAT as crucial intracellular signaling pathways in various biological processes such as cell survival, proliferation, differentiation, and inflammatory responses [56]. These pathways have shown importance in studies on hair follicle growth and development [57]. Akt promotes cell survival, inhibits apoptosis, and supports the growth phase of hair follicles [58]. Additionally, Akt influences the activity of hair follicle stem cells, promoting their proliferation and differentiation [59]. Akt activation is also linked to cellular metabolism and protein synthesis, which is essential for hair follicle formation [60]. STAT activation during the growth phase of hair follicles is crucial in regulating the hair follicle cycle [61]. For instance, STAT activation is associated with the transition of hair follicle stem cells from quiescence to growth phase. Furthermore, JAK inhibitors have been shown to affect the hair growth cycle, suggesting a potential role of the JAK/STAT signaling pathway in hair follicle growth and hair loss symptoms [62]. However, the involvement of these pathways in stress-induced hair loss remains unclear.

Immunofluorescence results from skin tissue sections of the various mouse groups indicate a significant decrease in the expression levels of p-PI3K, p-Akt, p-JAK, and p-STAT in the CUS group compared to the Control group. Upon treatment with SDF-1, the expression levels of p-PI3K, p-Akt, p-JAK, and p-STAT were significantly upregulated. This effect was reversed when blocking SDF-1 from binding to CXCR4 with AMD3100, with statistically significant differences observed (*p* < 0.05). Western blot analysis demonstrates a notable decrease in the expression levels of p-PI3K/PI3K, p-Akt/Akt, p-JAK/JAK, and p-STAT/STAT in the CUS group compared to the Control group. Following treatment with SDF-1, the expression levels of p-PI3K/PI3K, p-Akt/Akt, p-JAK/JAK, and p-STAT/STAT were significantly increased. Similarly, the blocking of SDF-1 and CXCR4 interaction with AMD3100 reversed these effects, with statistically significant differences (*p* < 0.05) (Figure 5A–C).

These results suggest that AMD3100 may effectively inhibit the binding of SDF-1 to the CXCR4 receptor, thereby regulating the activation of the PI3K/Akt and JAK/STAT pathways in hair follicles and influencing hair growth.

### 4.5. Silencing of the SDF-1 Gene Reduces p-PI3K/PI3K Expression and Inhibits Hair Follicle Cell Proliferation and Migration

In previous in vivo experiments, we elucidated the mechanism by which SDF-1 promotes hair growth through the PI3K/Akt and JAK/STAT signaling pathways. In this study, our aim was to understand the specific role of SDF-1 in hair follicle cell behavior by silencing SDF-1.

Initially, we examined the morphology of primary cells under bright-field microscopy and subsequently identified isolated and purified primary cells using Vimentin immunofluorescence staining (Figure 6A). Following this, we conducted Western blot and RT-qPCR experiments to observe the effect of SDF-1 knockdown. The results showed a significant decrease in SDF-1 protein and mRNA expression in the AMD3100 group compared to the Control group and a significant reduction in SDF-1 protein and mRNA expression in the siRNA-SDF-1 group compared to the NC-SDF-1 group, with statistically significant differences (*p* < 0.05) (Figure 6B). These results indicate that the present study successfully simulated downregulation of SDF-1 expression using siRNA in an ex vivo environment. Immunofluorescence results revealed a clear co-localization of SDF-1 and CXCR4 expression in both normal and SDF-1-knockdown hair follicle cells, indicating an interaction between these two proteins. The number of SDF-1^+^CXCR4^+^ cells was significantly reduced in the AMD3100 group compared to the Control group, as well as in the siRNA-SDF-1 group compared to the NC-SDF-1 group, with statistically significant differences (*p* < 0.05), suggesting that the decreased expression of SDF-1 affected its interaction with CXCR4 (Figure 6C). Western blot analysis showed that the levels of p-PI3K/PI3K were reduced in the AMD3100 group compared to the Control group, as well as in the siRNA-SDF-1 group compared to the NC-SDF-1 group, with statistically significant differences (*p* < 0.05) (Figure 6D). The cell wound healing test results demonstrated a significant decrease in cell migration 24 h post-treatment in the AMD3100 group compared to the Control group, as well as in the siRNA-SDF-1 group compared to the NC-SDF-1 group, with statistically significant differences (*p* < 0.05) (Figure 6E). Results from the CCK-8 assay showed a significantly reduced cell proliferation capacity in cells treated with AMD3100 compared to the Control group, as well as in cells treated with siRNA-SDF-1 compared to the NC-SDF-1 group, with statistically significant differences (*p* < 0.05) (Figure 6F), indicating the critical role of SDF-1 in cell proliferation and migration in the skin cells of chronic stress mice.

**Figure 5 biomolecules-14-01206-f005:**
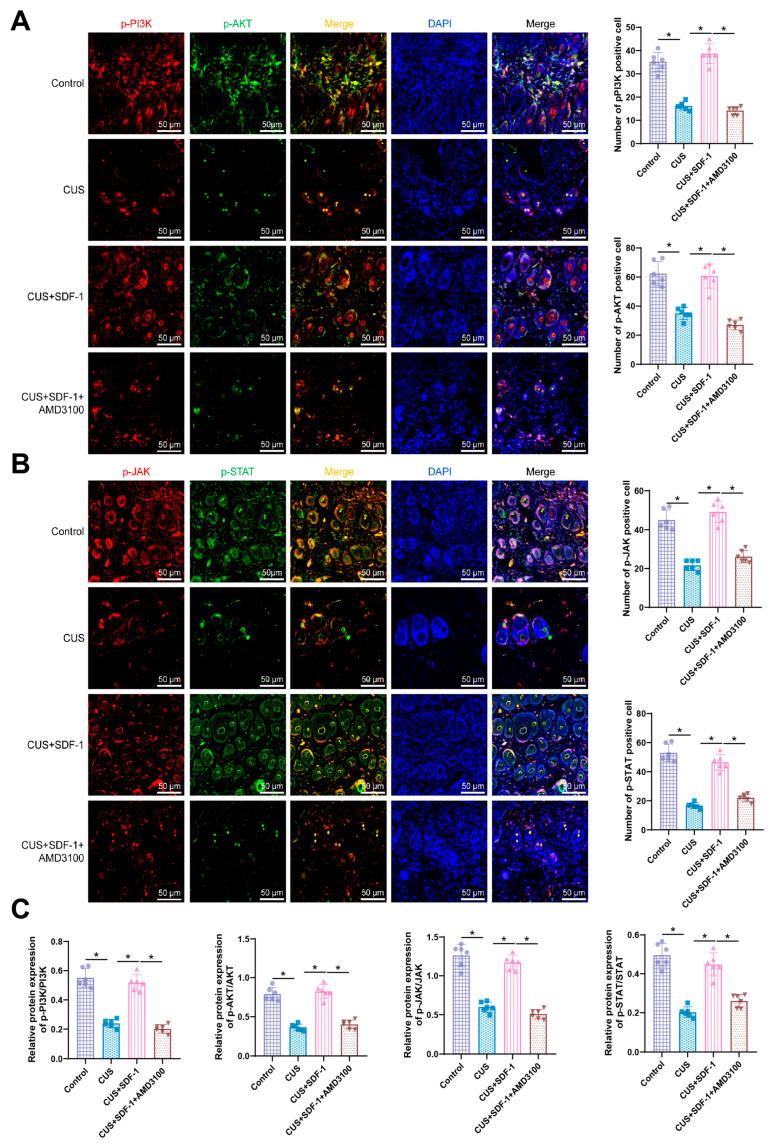
Impact of SDF-1 and AMD3100 on the protein expression of PI3K/Akt and JAK/STAT signaling pathways. Note: (**A**) Representative immunofluorescence images and statistical analysis of the PI3K/Akt signaling pathway in mouse skin tissue for each group, with a scale bar of 50 μm; (**B**) Representative immunofluorescence images and statistical analysis of the JAK/STAT signaling pathway in mouse skin tissue for each group; (**C**) Representative Western blot images and statistical analysis of the PI3K/Akt and JAK/STAT signaling pathways in mouse skin tissue for each group. Each group consisted of *n* = 6 mice; * indicates *p* < 0.05.

By genetically deleting SDF-1 in an in vitro model, we confirmed that SDF-1 regulates p-PI3K/PI3K and cell proliferation and migration through CXCR4. These results underscore the importance of SDF-1 in hair loss in CUS mice and provide a new direction for the discovery of novel clinical treatment methods in the future.

## 5. Discussion

In this study, we aimed to explore the mechanisms of the SDF-1/CXCR4 signaling axis in a mouse model of stress-induced hair loss. Through transcriptome sequencing and bioinformatics analysis, we found that under chronic stress conditions, SDF-1 may have a promoting effect on hair follicle growth, providing a new perspective on the pathogenesis of stress-induced hair loss. A key innovation of this study, in comparison to existing literature, lies in its focus on elucidating the regulatory role of the SDF-1/CXCR4 signaling pathway in stress-induced hair loss [16,17,63].

Previous studies have demonstrated the negative impact of stress factors on hair follicle growth, yet the mechanisms of action of the SDF-1/CXCR4 signaling axis in this process remain incompletely understood. This research experimentally confirmed the positive regulatory role of SDF-1, suggesting its significant role in promoting hair follicle growth. While building on previous research findings, this study expands the understanding of SDF-1 in stress-induced hair loss [17,64].

Moreover, we further investigated the regulatory effects of the SDF-1/CXCR4 signaling axis on the PI3K/Akt and JAK/STAT signaling pathways. The results indicate that SDF-1 may promote hair follicle growth by activating these two signaling pathways, offering new clues for studying the regulation mechanisms of hair follicle growth. Compared to the existing literature on these pathways, this study significantly elucidates their interplay under the regulation of SDF-1 [65].

Through mouse model experiments, we validated the potential of SDF-1 in treating stress-induced hair loss [66]. Experimental results revealed that subcutaneous injection of SDF-1 significantly enhanced hair follicle activity, while co-injection with the CXCR4 receptor inhibitor AMD3100 attenuated the promoting effect of SDF-1. These findings align with related studies and specify the role of the SDF-1/CXCR4 signaling axis in hair follicle growth [15,16,17].

Our research further reveals the regulatory mechanism of the SDF-1/CXCR4 signaling axis in the PI3K/Akt and JAK/STAT signaling pathways, providing new theoretical support for treating stress-induced hair loss. Exploring these breakthroughs will offer new avenues for future in-depth research. In harmony with previous studies on SDF-1/CXCR4, this study offers a more comprehensive explanation of the role of this signaling axis in hair follicle growth [16,67].

In summary, this study systematically investigates the role of the SDF-1/CXCR4 signaling axis in stress-induced hair loss in mice. Through 21 days of CUS treatment in C57 mice, we successfully established a model of chronic stress-induced hair loss and observed significant decreases in body weight, anxiety and depressive behaviors, as well as noticeable reductions in hair coverage and length (Figure 7). Transcriptome sequencing revealed 800 genes with significantly altered expression in skin samples of stressed mice compared to the control group, primarily enriched in biological processes related to stress response, inflammation, and cell migration. Weighted gene co-expression network analysis (WGCNA) and machine learning analysis identified a high correlation between the SDF-1 gene and chronic stress-induced hair loss, suggesting it may play a critical regulatory role in hair loss. Additionally, we found that the combined application of SDF-1 with the CXCR4 antagonist AMD3100 significantly promoted hair regeneration in stressed mice. Further experiments showed that SDF-1 could activate the PI3K/Akt and JAK/STAT signaling pathways of hair follicles while silencing the SDF-1 gene inhibited the proliferation and migration of hair follicle cells. In conclusion, these findings reveal the crucial role of the SDF-1/CXCR4 signaling axis in stress-induced hair loss in mice and provide an important molecular mechanism foundation for the development of future therapeutic strategies.

This study delves into the molecular mechanisms behind chronic stress-induced hair loss, highlighting the crucial role of the SDF-1/CXCR4 signaling axis in regulating hair follicle growth. Experimental validation demonstrates that upregulation of SDF-1 can promote hair follicle growth through the activation of the PI3K/Akt and JAK/STAT signaling pathways, offering novel targets and strategies for treating stress-induced hair loss. Clinically, this research could form the theoretical foundation for developing drugs targeting the SDF-1/CXCR4 signaling pathway, potentially offering more effective treatment options for patients with hair loss.

However, this study is subject to certain limitations, such as the use of only mouse models in the experimental design, without fully validating the results’ applicability in humans. Further research is needed to support the clinical prospects. Additionally, while the signaling pathways and molecular mechanisms investigated in this study are of significant importance, they require further in-depth exploration to elucidate their detailed regulatory mechanisms and interactions, ensuring the efficacy and safety of treatment strategies.

Future directions include expanding the sample size and optimizing experimental designs by incorporating more control groups and observations at different time points to verify the reliability and robustness of experimental results. Furthermore, further exploration of the role of the SDF-1/CXCR4 signaling axis in other types of hair loss and diseases can broaden the treatment scope. Coupling clinical observations with human experiments to unearth potential drug targets can accelerate the translation of this research into clinical practice, offering patients more effective and personalized treatment options.

## Figures and Tables

**Figure 1 biomolecules-14-01206-f001:**
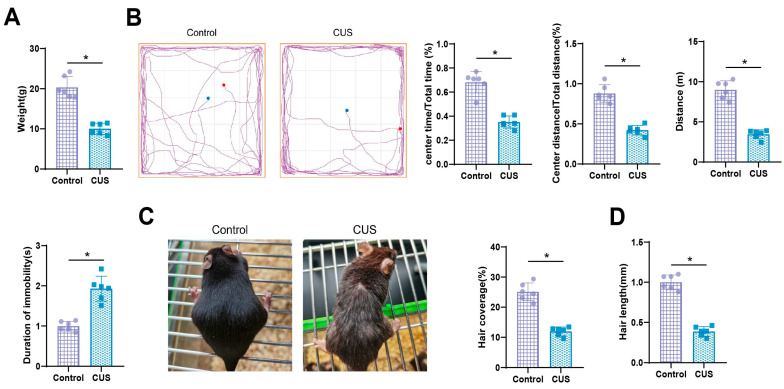
Effects of CUS on body weight, behavioral traits, and hair growth in mice. Note: (**A**) Comparison of body weight between two groups of mice; (**B**) Representative images and statistical graphs of open field and tail suspension tests for the two groups of mice; (**C**) Typical images and statistical graphs comparing hair coverage between the two groups of mice; (**D**) Comparison of hair length between the two groups of mice. Each group consisted of *n* = 6 mice; * indicates *p* < 0.05 compared to the Control group.

**Figure 2 biomolecules-14-01206-f002:**
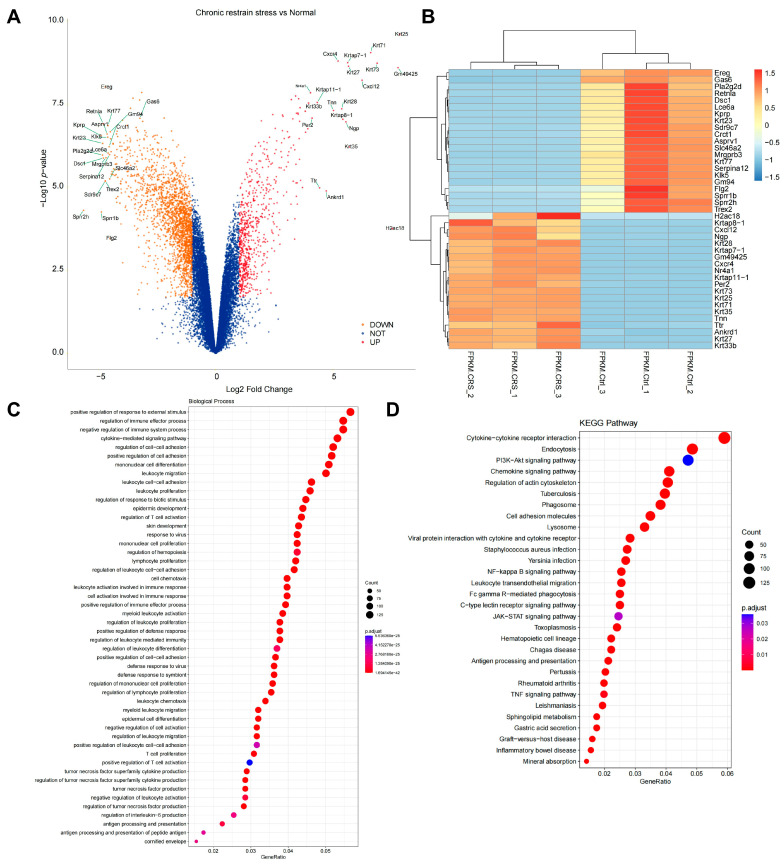
Assessment of chronic stress impact on gene expression and signaling pathway activity in mouse skin. Note: (**A**) Volcano plot demonstrating 2650 differentially expressed genes between the chronic stress group and the normal group of mice. Each point represents a gene, with position indicating the magnitude of expression change and statistical significance; (**B**) Heat map showing the most significant differentially expressed genes in mice under chronic stress; (**C**) Dot plot revealing enrichment of differentially expressed genes in biological processes such as stress response, inflammation, cell migration, and activation of inflammatory cells; (**D**) Enrichment analysis of KEGG pathways displayed in a dot plot showing the enrichment status of differentially expressed genes in various pathways, where the size of the dots represents the number of involved genes and the color intensity reflects the significance of enrichment.

**Figure 3 biomolecules-14-01206-f003:**
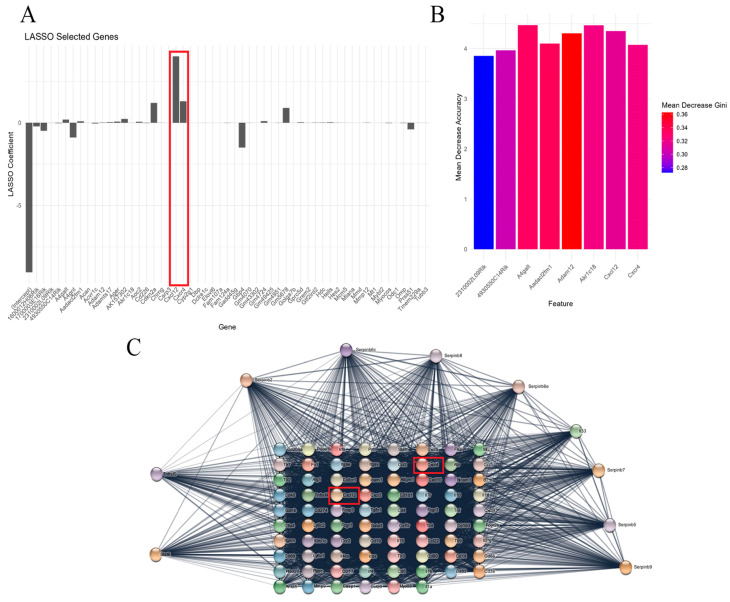
The key role of SDF−1 gene and its interaction network analysis under chronic stress. Note: (**A**) Genes highly correlated with chronic stress-induced hair loss selected through Lasso regression analysis; (**B**) Results from the random forest model demonstrating the high importance score of SDF−1/CXCR4 among all evaluated genes, further supporting its role as a critical regulatory factor in chronic stress-induced hair loss; (**C**) PPI network showcasing interactions between SDF−1/CXCR4 and multiple signaling pathway proteins, particularly with proteins interacting with CXCR4 receptor.

**Figure 4 biomolecules-14-01206-f004:**
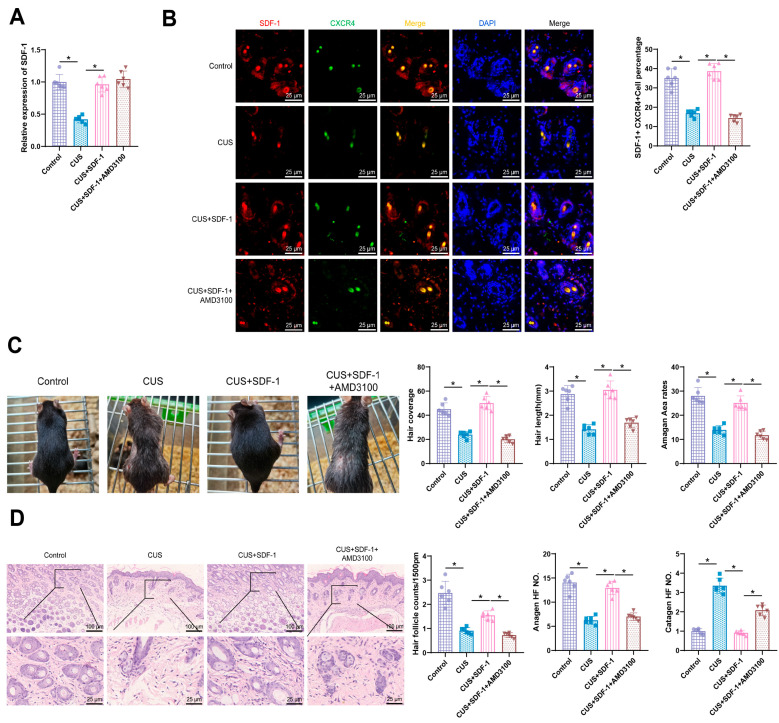
Impact of SDF-1 and AMD3100 on hair growth in CUS mice. Note: (**A**) ELISA analysis of SDF-1 concentration in the serum of mice in each group; (**B**) Immunofluorescence observation of the colocalization of SFD-1 and CXCR4 in mouse skin tissue (Scale bar: 25 μm); (**C**) Typical images and statistical graphs comparing hair coverage, hair length, and the proportion of skin area in the growth phase among the mouse groups; (**D**) Typical images and statistical graphs from H&E staining observation of the follicle count, proportion of follicles in the growth and regression phases among the mouse groups, labelled as 100/25 μm in the figures. Each group consisted of *n* = 6 mice; * indicates *p* < 0.05.

**Figure 6 biomolecules-14-01206-f006:**
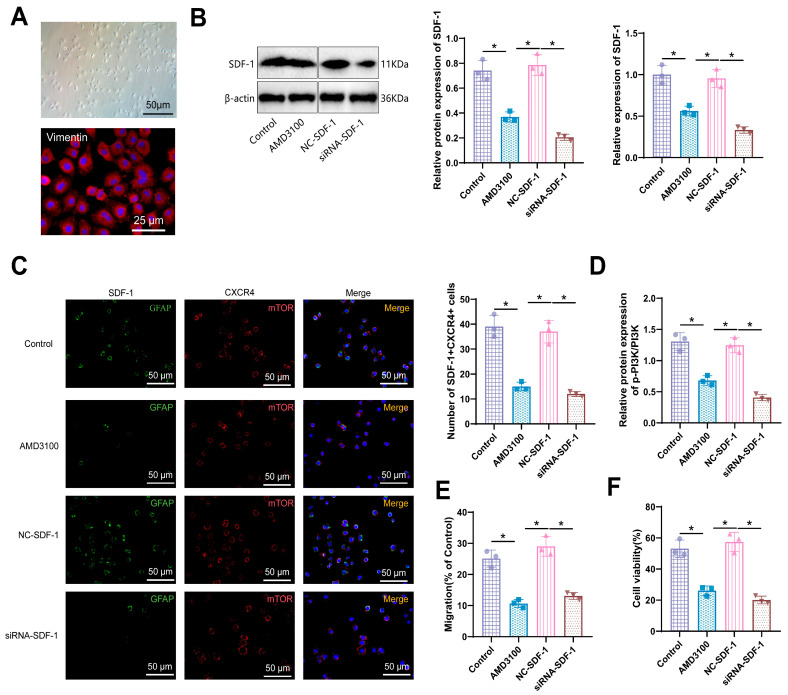
Effects of SDF-1 gene silencing on the expression and function of skin cells in mice under chronic stress. Note: (**A**) Bright-field microscopy (Scale bar: 50 μm) and immunofluorescence identification (Scale bar: 25 μm) of primary cell extraction and purification; (**B**) Western blot and RT-qPCR experiments examining the expression levels of SDF-1 mRNA and protein in mouse back skin cells after SDF-1 siRNA treatment. Original images can be found in Appendix A; (**C**) Immunofluorescence showing the colocalization of SDF-1 and CXCR4 in skin cells, with a scale bar of 50 μm; (**D**) Western blot analysis of p-PI3K/PI3K expression levels; (**E**) Representative images and statistical results of the cell wound healing test; (**F**) Assessment of cell proliferation using CCK-8 assay. * indicates significant differences between the two groups (*p* < 0.05), with all cell experiments repeated six times.

**Figure 7 biomolecules-14-01206-f007:**
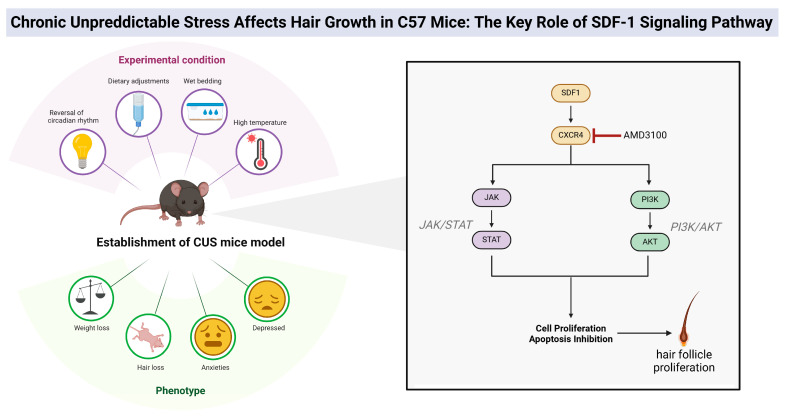
Molecular mechanisms of the SDF-1/CXCR4 signaling axis in chronic stress-induced hair loss in mice.

## Data Availability

The datasets generated and/or analysed during the current study are available in the manuscript and Appendix A.

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
