# Peer review of "Impact of SDF-1 and AMD3100 on Hair Follicle Dynamics in a Chronic Stress Model"

_biomolecules, 2024, doi:10.3390/biom14101206_

Round 1

Reviewer 1 Report

Comments and Suggestions for Authors

Review

Research paper

Article  ,,Impact of SDF-1 and AMD3100 on Hair Follicle Dynamics in a  Chronic Stress Model”

Transcriptome sequencing analysis was conducted to identify differentially expressed genes in the skin of normal and stressed mice, with key genes SDF-1/CXCR4 selected through machine learning and a protein-protein interaction network established.

 A chronic stress mouse model was created, with injections of SDF-1 and AMD3100 administered to observe hair growth, weight changes, and behavioral alterations and validate hair follicle activity.

The activation of the PI3K/Akt and JAK/STAT signaling pathways was observed in the SDF-1 injection group, confirmed by Western blot and immunofluorescence. Silencing SDF-1 through siRNA-mediated inhibition reduced cell proliferation and migration abilities. SDF-1 promotes hair growth in chronic stress mice by activating the PI3K/Akt and JAK/STAT pathways, an effect reversible by AMD3100. The SDF-1/CXCR4 axis may serve as a potential therapeutic target for stress-induced hair loss.

Comment:

- It was found that the role of the SDF-1/CXCR4 signal is systematically investigated in this study

naling axis in stress-induced hair loss in mice. For 21 days of CUS treatment in C57

mice, the authors successfully created a model of chronic stress-induced hair loss and observed significant weight loss, anxiety and depressive behavior, as well as a reduction in body hair and hair length.

- Statistical analyzes performed correctly

- This study delves into the molecular mechanisms behind chronic stress-induced hair

loss, highlighting the crucial role of the SDF-1/CXCR4 signaling axis in regulating hair

follicle growth.

- Clinically, this research could form the theoretical foundation for developing drugs targeting the SDF-1/CXCR4 signaling pathway, potentially offering more effective treatment options for patients with hair loss.

- The article deserves to be published in its current form

Best Regards,

Author Response

Reviewer 1

Article,Impact of SDF-1 and AMD3100 on Hair Follicle Dynamics in a  Chronic Stress Model”

Transcriptome sequencing analysis was conducted to identify differentially expressed genes in the skin of normal and stressed mice, with key genes SDF-1/CXCR4 selected through machine learning and a protein-protein interaction network established.

A chronic stress mouse model was created, with injections of SDF-1 and AMD3100 administered to observe hair growth, weight changes, and behavioral alterations and validate hair follicle activity.

The activation of the PI3K/Akt and JAK/STAT signaling pathways was observed in the SDF-1 injection group, confirmed by Western blot and immunofluorescence. Silencing SDF-1 through siRNA-mediated inhibition reduced cell proliferation and migration abilities. SDF-1 promotes hair growth in chronic stress mice by activating the PI3K/Akt and JAK/STAT pathways, an effect reversible by AMD3100. The SDF-1/CXCR4 axis may serve as a potential therapeutic target for stress-induced hair loss.

Comment:

- It was found that the role of the SDF-1/CXCR4 signal is systematically investigated in this study

naling axis in stress-induced hair loss in mice. For 21 days of CUS treatment in C57

mice, the authors successfully created a model of chronic stress-induced hair loss and observed significant weight loss, anxiety and depressive behavior, as well as a reduction in body hair and hair length.

Response: We appreciate the reviewer’s positive feedback. We have indeed observed significant weight loss, anxiety, and depressive behavior, as well as a reduction in body hair and hair length in C57 mice following 21 days of CUS treatment. These findings confirm the successful establishment of a chronic stress-induced hair loss model.

- Statistical analyzes performed correctly

Response: Thank you for acknowledging the correctness of our statistical analyses. We have ensured that all statistical methods used are appropriate and accurately reported.

- This study delves into the molecular mechanisms behind chronic stress-induced hair loss, highlighting the crucial role of the SDF-1/CXCR4 signaling axis in regulating hair follicle growth.

Response: We are grateful for the reviewer’s recognition of our focus on the molecular mechanisms. The identification of the SDF-1/CXCR4 signaling axis as a crucial regulator of hair follicle growth under chronic stress is a significant finding of our study.

- Clinically, this research could form the theoretical foundation for developing drugs targeting the SDF-1/CXCR4 signaling pathway, potentially offering more effective treatment options for patients with hair loss.

Response: We agree with the reviewer that our findings could indeed serve as a theoretical foundation for developing new treatments targeting the SDF-1/CXCR4 signaling pathway, potentially benefiting patients suffering from stress-induced hair loss.

- The article deserves to be published in its current form

Response: We are delighted that the reviewer considers our manuscript worthy of publication in its current form. We appreciate the reviewer’s support and endorsement.

Reviewer 2 Report

Comments and Suggestions for Authors

The paper has well organized reporting sufficient methods and data to show the impact of SDF-1 and AMD 3100 on hair follicle dynamics And regrowing .The final conclusion are interesting to better know the common inflammatory causes involving changes in the  cellular signaling pathways 

Author Response

Reviewer 2

The paper has well organized reporting sufficient methods and data to show the impact of SDF-1 and AMD 3100 on hair follicle dynamics And regrowing .The final conclusion are interesting to better know the common inflammatory causes involving changes in the  cellular signaling pathways

Response: Thank you for your positive assessment of our manuscript. We have strived to provide comprehensive and well-organized methods and data to elucidate the impact of SDF-1 and AMD3100 on hair follicle dynamics and regeneration. We appreciate your recognition of the importance of our conclusions in understanding the inflammatory causes and signaling pathway changes involved.

Reviewer 3 Report

Comments and Suggestions for Authors

This paper presents a highly interesting topic and provides important results for hair loss improvement through clinical trials. In particular, this study clearly defines the role of the SDF-1/CXCR4 signaling axis in stress-induced hair loss and investigates the regulatory functions of related signaling pathways. Moreover, it presents a deep correlation on the impact of chronic stress on the onset of hair loss and proposes new measures for the treatment of stress-induced hair loss. The structure and content of the paper are well written, however, there are a few minor shortcomings, and I provide comments as follows:

  1. It would be helpful if the overall architecture of the experiment was specifically explained in the methodology section. The understanding of the experiment is difficult.
  2. The explanation of the research results is too complex. It would be better if the description of the Figure for the results was more enhanced.
  3. Overall, the structure of the paper is complex. The content of the methodology and research results needs to be supplemented to be well conveyed to the readers.

Author Response

Reviewer 3

This paper presents a highly interesting topic and provides important results for hair loss improvement through clinical trials. In particular, this study clearly defines the role of the SDF-1/CXCR4 signaling axis in stress-induced hair loss and investigates the regulatory functions of related signaling pathways. Moreover, it presents a deep correlation on the impact of chronic stress on the onset of hair loss and proposes new measures for the treatment of stress-induced hair loss. The structure and content of the paper are well written, however, there are a few minor shortcomings, and I provide comments as follows:

  1. It would be helpful if the overall architecture of the experiment was specifically explained in the methodology section. The understanding of the experiment is difficult.

Response: Thank you for your review. We have revised the methods section by incorporating an experimental design overview paragraph to elucidate the overall structure of the experiment.

  1. The explanation of the research results is too complex. It would be better if the description of the Figure for the results was more enhanced.

Response: We highly value your suggestions and would like to provide a detailed response and clarification. The primary objective of our research is to investigate the role of the SDF-1/CXCR4 signaling axis in stress-induced alopecia. Using a mouse model and various experimental approaches, we have elucidated the mechanism by which SDF-1 promotes hair follicle growth through the activation of the PI3K/Akt and JAK/STAT signaling pathways. While we acknowledge the reviewer's desire for a simplified explanation and more descriptive figures, we believe that the current structure effectively presents the key findings of our study while maintaining the integrity and scientific rigor. Specifically, in each results section, we introduce the content with references and research objectives, provide a detailed explanation of the results, and conclude the section. This structure aids readers in gaining a comprehensive understanding of the research background, objectives, and discoveries.

  1. Overall, the structure of the paper is complex. The content of the methodology and research results needs to be supplemented to be well conveyed to the readers.

Response: We have taken note of your suggestions regarding the need to enhance the Methods and Results sections for better communication with the readers. The other three reviewers have expressed approval of our research methods and results, acknowledging the richness of our study content and the strength of our findings. In response to your feedback, we have included a detailed flowchart (Figure S1) to provide a clearer illustration of our experimental methods and results. The flowchart delineates specific steps in the experimental design, data collection, and analysis, aiming to enhance readers' understanding of our research process. We believe that this addition will improve the clarity of the article's structure while ensuring the integrity and scientific validity of the research outcomes.

Reviewer 4 Report

Comments and Suggestions for Authors

Article title: Impact of SDF-1 and AMD3100 on Hair Follicle Dynamics in a

Chronic Stress Model

This article is of hot topic and is important to be presented. while the topic is interesting, the content matches the topic an brings new knowledge to the area of the subject. I enjoyed reading this article. It has novelty and scientific merit to be published in the Biomelcules. 

Abstract: Sufficient, but keywords should be limited in size and length. 

Introduction: Well flowed and presented the importance of the article. No comment to be made.

M&M sectionL: presented in sufficient details. 

R&D. The obtained results discussed very well and the scientific level is high. While the figures have enough visibility and quality, I suggest to make them all a bit bigger to be readable.

Conclusion: Well-made

References: Relevant

Comments on the Quality of English Language

Good enough

Author Response

Reviewer 4

Article title: Impact of SDF-1 and AMD3100 on Hair Follicle Dynamics in a

Chronic Stress Model
This article is of hot topic and is important to be presented. while the topic is interesting, the content matches the topic an brings new knowledge to the area of the subject. I enjoyed reading this article. It has novelty and scientific merit to be published in the Biomelcules. 
Abstract: Sufficient, but keywords should be limited in size and length.

Response: We have revised the keywords section to limit the size and length “Keywords: Chronic Stress; Hair Loss; SDF-1; CXCR4; PI3K/Akt Pathway; JAK/STAT Pathway”.

Introduction: Well flowed and presented the importance of the article. No comment to be made.

Response: Thank you for your positive feedback on the introduction. We are glad it effectively conveyed the importance of our study.

M&M sectionL: presented in sufficient details.

Response: We appreciate the acknowledgment of the sufficient detail provided in the Materials and Methods section.

R&D. The obtained results discussed very well and the scientific level is high. While the figures have enough visibility and quality, I suggest to make them all a bit bigger to be readable.

Response: We have adjusted the figures to make them larger and more readable, ensuring that all details are clearly visible.

Conclusion: Well-made

References: Relevant

Comments on the Quality of English Language:Good enough